# Machine Learning-Based Prediction of Specific Energy Consumption for Cut-Off Grinding

**DOI:** 10.3390/s22197152

**Published:** 2022-09-21

**Authors:** Muhammad Rizwan Awan, Hernán A. González Rojas, Saqib Hameed, Fahid Riaz, Shahzaib Hamid, Abrar Hussain

**Affiliations:** 1Department of Mechanical Engineering, Universitat Politecnica De Catalunya (UPC), 08034 Barcelona, Spain; 2Department of Mechanical Engineering, The Superior University, Lahore 54000, Pakistan; 3Department of Mechanical Engineering, Abu Dhabi University, Abu Dhabi P.O. Box 59911, United Arab Emirates; 4Department of Research and Development, AI Studio, Lahore 54000, Pakistan; 5Department of Mechanical and Industrial Engineering, Tallinn University of Technology, Ehitajate Tee 5, 19086 Tallinn, Estonia

**Keywords:** artificial intelligence, machine learning, energy prediction, data-driven modeling, advanced manufacturing, intelligent grinding

## Abstract

Cut-off operation is widely used in the manufacturing industry and is highly energy-intensive. Prediction of specific energy consumption (SEC) using data-driven models is a promising means to understand, analyze and reduce energy consumption for cut-off grinding. The present article aims to put forth a novel methodology to predict and validate the specific energy consumption for cut-off grinding of oxygen-free copper (OFC–C10100) using supervised machine learning techniques. State-of-the-art experimental setup was designed to perform the abrasive cutting of the material at various cutting conditions. First, energy consumption values were predicted on the bases of input process parameters of feed rate, cutting thickness, and cutting tool type using the three supervised learning techniques of Gaussian process regression, regression trees, and artificial neural network (ANN). Among the three algorithms, Gaussian process regression performance was found to be superior, with minimum errors during validation and testing. The predicted values of energy consumption were then exploited to evaluate the specific energy consumption (SEC), which turned out to be highly accurate, with a correlation coefficient of 0.98. The relationship of the predicted specific energy consumption (SEC) with material removal rate agrees well with the relationship depicted in physical models, which further validates the accuracy of the prediction models.

## 1. Introduction

Industrial manufacturing is one of the main contributors to global warming, with a 30% share of primary energy consumption [1]. To minimize the environmental impact of the manufacturing processes, reduction of energy consumption is the priority of manufacturers [2]. Among the manufacturing processes, grinding, being a multipoint, cutting operation, is a complex and highly energy-intensive process [3]. Understanding and estimation are primary and important steps to reduce the energy consumption of machining [4]. Specific energy consumption (SEC), i.e., the energy required to remove the unit volume of the material, is an important indicator to understand and analyze the energy consumption pattern among different materials [5]. The abrasive cut-off operation, which is the thin sectioning of the metals with cutting discs, is a high-material-removal-rate grinding process and consumes a lot of energy [6,7]. Therefore, specific energy prediction and analysis for abrasive cut-off operation are of paramount importance. The specific energy of grinding and its associated components are highly influenced by the process parameters [8]. Numerous experimental and modeling techniques have been directed to model and evaluate grinding energy and its components [5,9,10,11]. However, these techniques rely on complex mathematical formulations to find the relationship between the grinding energy and the process parameters and are based on specific assumptions, which might vary during the experimental conditions. 

For this reason, the data-driven approach based on machine learning is the focus of the researcher because of its ability to find highly complex and nonlinear patterns in data and transform raw data into models for prediction, detection, classification, and regression [12]. Several studies have been conducted to predict the energy consumption of different machining processes based on machine learning techniques. Kant [13] developed the artificial neural network model to predict the cutting energy of carbon steel during machining using data from 27 experiments by varying process parameters (spindle speed, depth, and width of cut and feed rate). The employed model predicted the cutting energy with almost 98.5% accuracy. Similarly, Borgia et al. [14] predicted the energy consumption of milling using an artificial neural network; the employed model was able to predict the energy consumption of machining with an error of 2.46%. Bhinge et al. [15] employed a Gaussian process-based data-driven approach to predict the energy consumption function for machine tools that can be generalized over multiple process parameters and operations. Liu et al. [16] introduced the hybrid technique based on Gaussian process regression and process mechanics to predict the specific cutting energy of milling. Brillinger et al. [17] employed the machine learning algorithms of decision tree, random forest, and boosted random forest to predict the energy consumption of CNC based on real production data. Their research results revealed that random forest is the most accurate algorithm for energy prediction. Ziye and Yue bin [16] presented the hybrid approach to predict the specific cutting energy of a milling machine by integrating the data-driven machine learning model and process mechanics. Stefanio et al. [14] used ANN to analyze the influence of process parameters on the energy consumption of milling process. Similarly, Quintana et al. [2] optimized the power consumption of the milling process using ANN and determined the most effective process parameters to save energy and process cost.

Similar to other machining processes, data-driven conditioning monitoring and prediction of indicators for grinding are also being extensively applied. For instance, Siamak et al. [18] predicted the surface roughness and cutting forces of grinding using process parameters and acoustic emissions signals (AE) as the input. Predicted results were in good agreement with experimental data with an accuracy of 99%. Sauter et al. [19]. employed the multiclass classification using supervised learning to predict the grinding burn based on input process measurements of acoustic emissions, spindle electric current, and power signals. Arriandiaga et al. [20] presented an ANN-based approach to predict and model the specific energy consumption of grinding using wheel characteristics and the grinding conditions [20]. Similarly, He et al. [21] developed a generic energy prediction model of machine tools using deep learning algorithms. He employed unsupervised machine learning to extract sensitive features from raw data of milling and grinding machines and then used the supervised machine learning algorithms to develop a prediction model between the extracted features and the energy consumption of machine tools.

Unlike the comprehensive studies of machine learning-based energy prediction of machining and surface grinding, none of the efforts have been directed towards machine learning or data-driven-based prediction of specific energy consumption for cut-off grinding, which is the widely used finishing operation in the industry. The recently conducted and previous studies about the abrasive cut-off operation for metals, stone machining, and concrete rely on the complex modeling of specific energy consumption in relationship with the process parameters [5,22,23,24,25]. These models are true under certain assumptions and for specific conditions and might not represent an accurate estimation of energy during data variation due to practical constraints. To fill this gap, this research aims to present a novel data-driven approach for the prediction of specific energy consumption of cut-off grinding using supervised machine learning techniques of decision trees, Gaussian process regression, and ANN. A state-of-the-art grinding machine was developed and a novel procedure was designed to perform abrasive cutting on oxygen-free copper under different cutting conditions. The developed algorithms showed very good prediction accuracy when explored and tested for the new data set. 

## 2. Methodology

A novel methodology has been adopted to predict and validate the specific energy consumption of abrasive metal cutting with thin discs.

The specific energy consumption is defined by the relationship:(1)$$SEC=\frac{P_{m}}{Q_{w}}$$

The prediction of specific energy consumption (*SEC*) involves measurement and prediction of both cutting power (*P_m_*) and material removal rate (*Q_w_*). Figure 1 elaborates the overall process, from experimentation to modeling and then prediction of specific energy consumption. First, abrasive metal cutting procedure is employed, as shown in Figure 2, to measure the electrical energy consumed during the process. The data of energy consumption (*E_e_*) is collected, cleaned, and processed to apply machine learning algorithms, which provide the prediction of *E_e_* based on input process parameters. The predicted energy is used to evaluate the mechanical cutting power (*P_m_*) from the experimental relationship of *E_e_* and *P_m_* obtained through the calibration of standard grinder with dynamometer. Based on the predicted values of cutting power (*P_m_*) and material removal rate (*Q_w_*), specific energy consumption is predicted and compared with experimental values. 

### 2.1. Design of Machine and Experimental Settings

To measure the electrical energy consumed (*E_e_*) for abrasive cutting of oxygen-free copper, a state-of-the-art automated experimental setup was designed and developed. This experimental set up was automated through Arduino Uno. The abrasive metal cutting was performed by 1 × 115 mm cutting discs, which were mounted on the standard grinder. The machine is capable of cutting different material at four predefined feed rates. The downfeed movement of the machine was controlled through stepper motor, which was programmed in Arduino.

The stepper motor drive TB 6600 was used to control the current between stepper motor and control circuit in microsteps. The feed rate range can be altered by varying the number of steps of the stepper motor. The detailed Arduinio program is given in Figure 3 and Figure 4. 

Stepper motor is preferred due to machine requirements for higher accuracy in speed, torque control, and dynamic loading [26]. A switch relay was used to control the grinder buttons, emergency stop buttons, and limit switches.

Four LED lights were the indicators of feed rates, and were controlled by the potentiometer “P”. Up and down movements of the machine were controlled by two vertical switches S1 and S2, through stepper motor. These switches respond according to the feed rate values decided by the potentiometer. These switches move the stepper motor in clockwise or anticlockwise direction to move the machine up and down through a lead screw, which converts the rotary motion into linear motion. Limit switches were used to restrict the up and down movement of the machine within controlled limits and were also programmed with microcontroller Arduino Uno.

As shown in Figure 3 and Figure 4, potentiometer was assigned with different range of values of the LEDs, and for each LED a particular value of the velocity was defined. LED value range is defined between (0–1024), along with velocity range for four velocities between (100–500). The function first reads the particular value of the LED; if it is high and other values are low, it executes the particular velocity and reads the switch 1 (S1) and switch (S2) states. 

If switch (S1) is on, which is also defined by the particular LED 5, the function will execute the velocity. Stepper motor being turned on, high in the flow chart, means that the machine will move up. During the execution state, if the limit switch is pressed, the program will turn off the stepper motor. If switch (SW2) is on, which is indicated by LED6, the stepper motor will be turned on and the machine will move in the downward direction, and if the limit switch is pressed, the program will turn off the stepper motor. In the flow chart, Vel microseconds delay indicates that the motor moves very little between microsteps, and the waiting time between the steps is reduced to achieve the smooth function of the motor. The energy consumed during this process is processed through a power analyzer and then stored in the computer through a channel recorder. These energy consumption values are used to calculate the cutting power (*P_m_*) through the relationship defined by
(2)$$P_{m}=2.4\times E_{e}-742.61$$

The detailed procedure for energy consumption and the corresponding cutting power measurement is elaborated in the previous work [5].

### 2.2. Cutting Conditions

The experimental cutting conditions are shown in Table 1. The abrasive cutting action was performed on oxygen-free copper (OFC–C10100) having a thickness of 10.8 mm. Cubitron and standard steel cutting discs were used to cut specimens at predefined feed rates of 0.538, 0.639, 0.899, and 1.488 mm/s. A 115 mm, 660 W Ryboi grinder with a maximum speed of 13,000 rpm [27], was used to mount the cutting discs of 1 mm thickness, 22 mm bore diameter, and 115 mm outer diameter. To analyze the influence of cutting grits on specific energy consumption, two different cutting tools were used [28,29]. The detailed technical characteristics of the cutting discs are specified by the code, which indicates, grit size, type, bonding material, and abrasive material. The cubitron cutting discs are rigid and have reinforced bonded wheels. The cutting tools are made up of precision-shaped abrasive grains, which are self-sharpening and have higher cutting efficiency [30]. The detailed data sets for these cutting conditions are given in Appendix A. 

### 2.3. Data Acquisition

The energy consumed during the abrasive cutting was measured with a power analyzer built with an AD633 chip and stored in the computer through channel recorder logger K8047. Power signals were collected with sampling frequency of 0.01 s for four feed rate values of 0.538 mm/s, 0.639 mm/s, 0.899 mm/s, and 1.488 mm/s. Hysteresis dynamometer (HD-710-BNA) was used for the calibration of standard grinder to measure the relationship between electrical energy consumed (*E_e_*) and mechanical cutting power (*P_m_*). 

A high-speed programmable controller attached with a dynamometer measured torque, velocity, current, voltage, cutting power (*P_m_*), and electrical energy consumed (*E_e_*). The relationship between mechanical cutting power (*P_m_)* and electrical energy consumed (*E_e_*) as shown in Figure 5 is linear. This regression equation was used to find the values of mechanical cutting power (*P_m_*) during experimentation, shown in Figure 2. 

A digital microscope [WADEO iT33-MDUK] was used to measure the cutting thickness (a), in Figure 6 in millimeters at different points of the cutting path, and was then analyzed through Image J2 open source software. The digital microscope can measure the variation in cutting thickness with an accuracy of 0.001 mm. The width of the specimen (b), as shown in Figure 6, was also measured in millimeters with a digital vernier caliper. Figure 7 shows the energy consumption behavior of oxygen-free copper (OFC–C10100) at a feed rate of 0.538 mm/s with standard steel cutting disc Inox. Point A to B is the free-cutting condition, as the disc does not make contact with the specimen. 

Between points B and D, the cutting disc engages with the workpiece, and the energy consumption starts to increase. 

At point D, energy consumption usually stabilizes. The average value of the energy consumed during this stabilization stage is used to measure the energy consumption [5,31]. However, in this particular case, it can be seen that energy did not stabilize for the whole time period between D and G. It stabilizes in two stages, at E and F, and the average values of the energy consumed at both of these points were used for evaluation purposes. This behavior is attributed to both the nonuniform structure of the material and the vibration of the cutting disc at a low material removal rate. Due to this vibration of the cutting disc and non-homogeneity of the material, the cutting thickness, as shown in Figure 6, also varies, which causes a variation in material removal rate. Therefore, at the same feed rate, it is possible to have slightly varied values of energy consumption and material removal rate. In this study, these varied values of the energy consumption and material removal rate at the same cutting conditions provide the opportunity to have more realistic data points for the training of the machine learning models.

At higher feed rates, the vibration of the cutting disc reduces, which reduces the variation of energy consumption and cutting thickness of the specimen. The vibration of the cutting disc was also found to be affected by the cutting tool. In this study, cutting with Cubitron cutting disc was found to be very precise with minimum variation. This is attributed to the precise triangular abrasive grits of the Cubitron cutting disc, which reduces the vibrations and increases the cutting efficiency [32]. 

The material removal rate (*Q_w_*) is defined by the product of feed rate (*V_f_*) and cutting cross section (*A_c_*) [3].
(3)$$Q_{w}=A_{c}\times V_{f}$$
$$A_{c}=a\times b$$
where *V_f_* is a function of the speed of the stepper motor and *A_c_* is the product of the cutting thickness and width of the specimen, as shown in Figure 6 [5]. The downward feed speed in relation to stepper motor speed was determined by conducting the kinematic study using generalized coordinates [33]. It was revealed that at a constant rotation speed of the stepper motor, feed speed was perpendicular in relation to the workpiece, and remained constant throughout the cutting cross section.

### 2.4. Data Processing and Machine Learning Models

The experimental data of energy consumption (*E_e_*) obtained at different feed rates, cutting velocities, and cutting tools were processed to select the most relevant features. The ugly data app of Matlab was selected to remove the anomaly at the threshold factor of 2 from the mean data, and the moving median was used to fill the missing data. 

After data processing, the regression learner app was used to apply supervised machine learning. Regression trees, Gaussian process regression, and artificial neural networks were used to predict specific energy consumption.

#### 2.4.1. Regression Trees

A regression tree develops the regression model in the form of the tree structure. Data sets are broken down into smaller and smaller data sets, while at the same time incrementally constituting the associated decision tree. The final result is a tree with a decision node and a leaf node, where the leaf represents the outcome and decision nodes are the points where the data are split [34]. Fine regression trees were used with a minimum leaf size of 4.0 and cross-validation folds of 5.0.

#### 2.4.2. Gaussian Process Regression

Gaussian process regression (GPR) is developed based on the assumption that output $y=f_{q}\left(x\right)+\epsilon$ is measured with noise $\in \sim N\left(0,\sigma_{\epsilon \;}^{2}\right)$, which is also Gaussian-distributed with zero mean and variance $\sigma_{\epsilon \;}^{2}$. In GP, values of unknown function $f_{q}\left(x\right)$ are treated as random variables and modeled by Gaussian distribution for incorporating previous knowledge acquired in the historical data [15]. Assuming that $D=\left\{\left(x_{i},y_{i}\right),\;i=1,2,\ldots \ldots \ldots ,\;N\right\}$ represents the training data set of the Gaussian model. The feature vectors $x_{i}\in R^{N}$ consist of extracted features and the corresponding process parameters, for instance, feed rate, cutting tool, cutting thickness, etc. The observed target value $y_{i}$ represents the energy consumption (*E_e_*) during the cut-off grinding, so $X={x_{i}}_{i=1}^{N}$ represents the input matrix of the training data set, and $y={y_{i}}_{i=1}^{N}$ represents the output vector. The Gaussian process is defined by its mean function m(x) and its covariance function $k\left(x,x^{\prime }\right)$ [35]. The Gaussian process is represented as
(4)$$f\left(x\right)\sim GP\left(m\left(x\right),k\left(x,x^{\prime }\right)\right)$$
where m(x)=E[f(x)]
(5)$$k\left(x,x^{\prime }\right)=E\left[f\left(x\right)-m\left(x\right)\right)\left(f\left(x^{\prime }\right)-m\left(x^{\prime }\right)\right)]$$

The mean function m(x) showed the expected value of the function f(x) corresponding to input *x*. The covariance function can be considered as the measure of the confidence level for m(x) [36]. The training was performed at a constant basis function, with a rational quadratic kernel function. 

#### 2.4.3. Artificial Neural network

The artificial neural network (ANN) constitutes a data processing system, which is inspired by the human biological neural network. Experimental or analytical data sets can be used in ANN to model the behavior of the system with different influencing factors. It consists of an input layer, one or more hidden layers, and the output layer. Neuron is considered as the elementary unit in each layer [37]. The supervised learning principle for neural networks was considered in this model. In supervised learning, inputs and correct outputs are provided to the network, which processes and compares the results to the desired outputs. The errors, which are the differences between desired output and network output, are backpropagated, and weights are adjusted accordingly. The ANN architecture was trained using two hidden layers, with ReLU as the activation function.

#### 2.4.4. Performance Metrics

The performance of regression models was determined using indicators of root-mean-square error (RMSE), determination coefficient R, mean-square error (MSE), and mean absolute error (MAE) [21]. These values are determined through the following relationships:(6)$$\;\;\;\;\;\;\;\;\;\;\;\;\;\;\;RMSE=(\left(\frac{1}{N}\right)\left(\sum_{i=1}^{N}{\left|t_{i}-y_{i}\right|}^{2}\right){)}^{1/2}$$
(7)$$R^{2}=1-\left(\frac{\sum_{i=1}^{N}{\left|t_{i}-y_{i}\right|}^{2}}{\sum_{i=1}^{N}y_{i}{}^{2}}\right)$$
(8)$$MSE=\frac{1}{N}\sum_{i=1}^{N}{\left|t_{i}-y_{i}\right|}^{2}{}^{}$$
(9)$$\;\;\;\;\;\;\;\;\;\;\;\;\;\;\;\;\;\;\;\;\;\;\;\;\;MAE=\frac{1}{N}\sum_{i=1}^{N}{\left(t_{i}-y_{i}\right)}^{}$$
where *N* is the number of samples, *t_i_* is the target value, and *y_i_* is the predicted value.

## 3. Results and Discussion

The predicted values of electrical energy consumption, which were trained with input process parameters, are shown in Figure 8. 

These figures show the results of about 130 measurements based on approximately 35 experiments to evaluate energy consumption (*E_e_*) based on process parameters. The general increasing trend of energy consumption is in agreement with the previous research on grinding [5]. The results of selected features for validation and testing for Gaussian process regression (GPR), regression trees, and artificial neural network (ANN) are shown in Figure 8. Figure 9 shows the true vs predicted response of the adopted model.

To protect the data against overfitting, it was partitioned into folds, and accuracy was estimated on each fold. A cross-validation fold value of 5.0 was selected for these data sets. These values are obtained by hit-and-trial experiments by tuning the hyperparameters of the chosen models. It can be seen that GPR achieved the best accuracy in validation and testing, with R^2^ values of 0.78 and 0.84, respectively. After GPR, the ANN performance was better, with an R^2^ value of 0.73 in validation; however, in testing, ANN performed worst among all three models. The computation cost was measured in terms of the training time of the algorithm to predict the data. The regression trees were superior in terms of training time with 0.358 s, while computation of ANN took 3.508 s, followed by GPR with the highest computation time of 4.259 s. Despite the high computation time, it can be deduced that GPR’s performance is superior both in training and testing. 

To improve the performance of the existing models, hyperparameter optimization was performed by using Bayesian optimization with the objective of minimizing the cost function of MSE, as shown in Figure 10 and Figure 11. The data set for this experiment is small, so the default number of iterations and time limit is used. The blue line shows the minimum MSE calculated by the optimization process, considering all the hyperparameter values. The dark line represents the observed minimum MSE at a particular time; for instance, at a fifth iteration, the dark line will show the minimum observed MSE in the first five iterations.

The red dot indicates the best value of iteration at which hyperparameters are optimized. In this particular experiment, at the best hyperparameter points, the minimum MSE values were 35.02 for Gaussian process regression and 48.8 for regression trees. The optimization of the adopted models indicates the accuracy of results obtained earlier, as GPR performs better with lower values of minimum MSE. Therefore, based on the optimization results, it can be deduced that for this data set, Gaussian process regression (GPR) is the best model. To evaluate the accuracy of these data-driven models, predicted values of energy consumption (*E_e_*) were used to further predict the cutting power (*P_m_*) through an equation that was obtained by calibrating the standard grinder with the dynamometer. 

Using this predicted cutting power (*P_m_*) and predicted values of material removal rate, specific energy consumption (SEC) was predicted and validated. Figure 12 shows the validation of energy consumption values in terms of evaluation of specific energy consumption. The high value of the coefficient of determination R^2^ for predicted specific energy consumption (SEC) indicates that the chosen models, even with moderate performance (R^2^ values between 0.7–0.78), can be good enough to accurately predict the associated values of energy consumption, such as cutting power, specific energy consumption, etc. The specific energy consumption decreases with an increase in material removal rate; this behavior is similar to the previous research on grinding [5,38,39] and also validates the accuracy of predictions. It can be seen that at a low material removal rate, the predicted values of the three algorithms are scattered, but as the material removal rate increases, values become closer to each other.

This behavior is attributed to the large vibration of the cutting disc at a lower feed rate, due to which the energy consumption values vary significantly, so values are more scattered. As the feed rate increases, the vibration of the cutting disc reduces, and at higher feed rates, the vibration of the cutting disc is minimum. Therefore, the original and predicted values are closer to each other. The training of these machine learning models for other materials such as steel, aluminum, and intermetallics, and the adopted methodology make a very good application for the calibration of the industrial grinder and cutting discs in terms of specific energy consumption, cutting power, and material removal rate.

## 4. Conclusions

A state-of-the-art machine was designed to automate the movements of a standard manual grinder that is widely used in the industry. The equipment allowed for the precise measurement of the specific energy consumption of cutting discs using a standard grinder for oxygen-free copper at predefined cutting conditions. A combination of a dynamometer and a smart cut-off grinding machine for the experimental data collection of cutting power and material removal rate presents a new method to measure and evaluate the specific energy consumption. Three supervised machine learning techniques of ANN, GPR, and regression trees were employed to predict the specific energy consumption. Gaussian process regression performed better in validation and testing as compared to ANN and regression trees. Hyperparameter optimization was performed to minimize the MSE, which also verified the GPR as the best model for this data set among the tested models. To evaluate the accuracy of the machine learning models, predicted energy consumption values were applied to calculate the cutting power and specific energy consumption. With regard to exploitation, high correlation coefficient values (R > 0.98) between the experimental SEC and the predicted SEC of three machine learning models indicate that predicted energy consumption (Pe) values with even moderate performance are reliable to accurately predict specific energy consumption. Moreover, the relationship of the predicted SEC with material removal rate shows the same trend as depicted by physical models [5,24], which also reinforces the accuracy of the prediction models. Machine learning-based prediction of energy consumption of widely used cutting discs introduces a new approach to calibrate the standard grinders and cutting discs in terms of cutting power, material removal rate, and specific energy consumption. The accuracy of the adopted models can be further improved by improving the data set. In the future, these machine learning models can be trained on large data sets to include other materials and cutting conditions, which will make it a very good application to predict the energy consumption of cutting discs used in manufacturing, stone machining, and the concrete industry. 

## Figures and Tables

**Figure 1 sensors-22-07152-f001:**
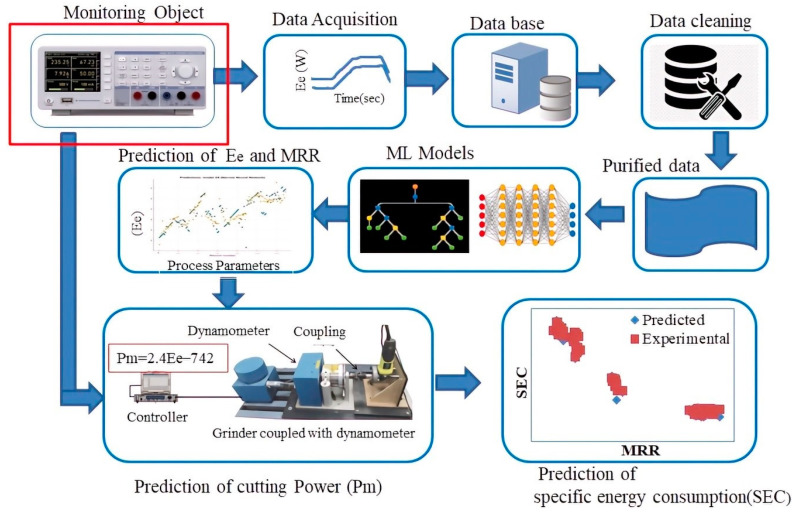
Overall procedure for prediction of specific energy consumption (SEC).

**Figure 2 sensors-22-07152-f002:**
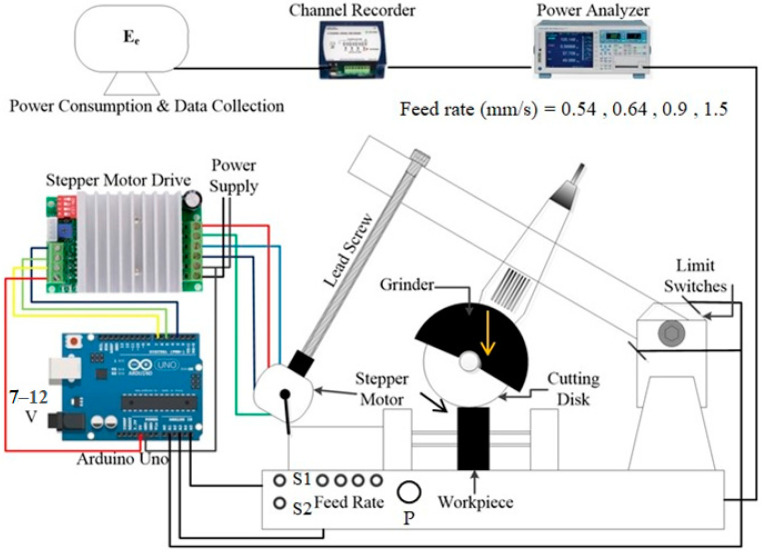
Machine for cut-off grinding of oxygen-free copper.

**Figure 3 sensors-22-07152-f003:**
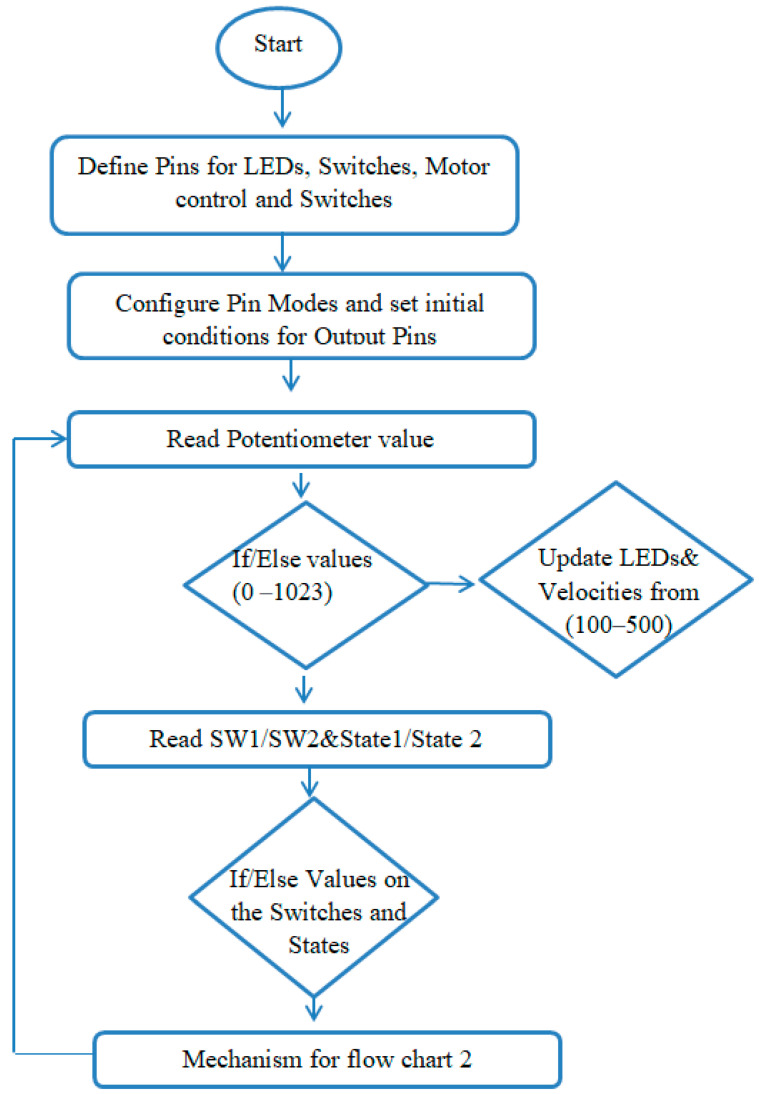
Flow chart of the Arduinio program to control the movements of machines through stepper motor.

**Figure 4 sensors-22-07152-f004:**
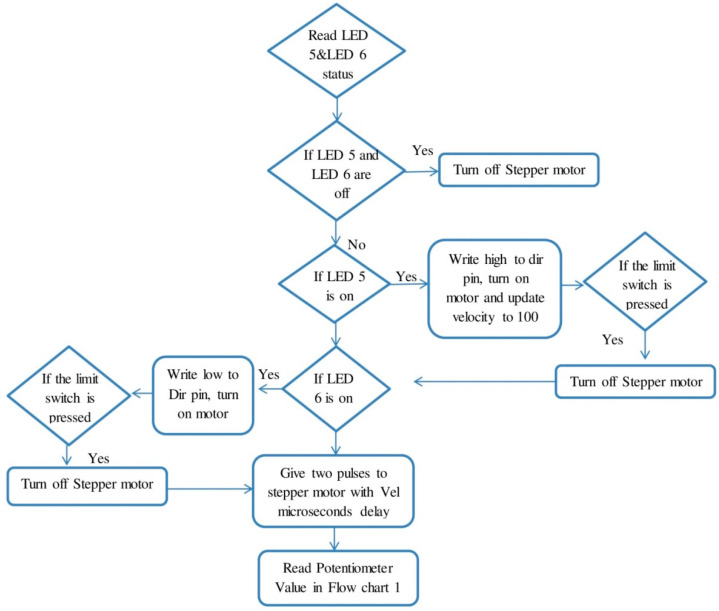
Flow chart of the Arduino program to control the movements of machines through stepper motor.

**Figure 5 sensors-22-07152-f005:**
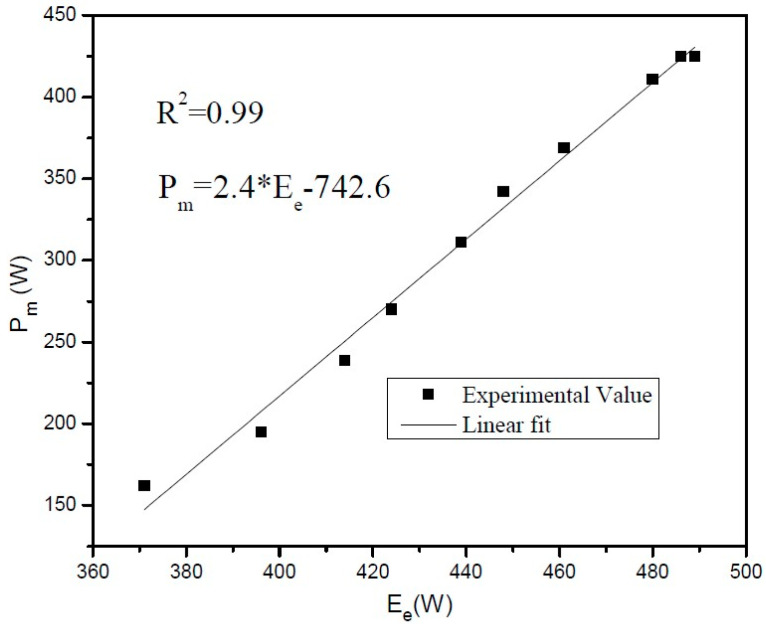
Relationship between electrical energy (*E_e_*) and mechanical cutting power (*P_m_*).

**Figure 6 sensors-22-07152-f006:**
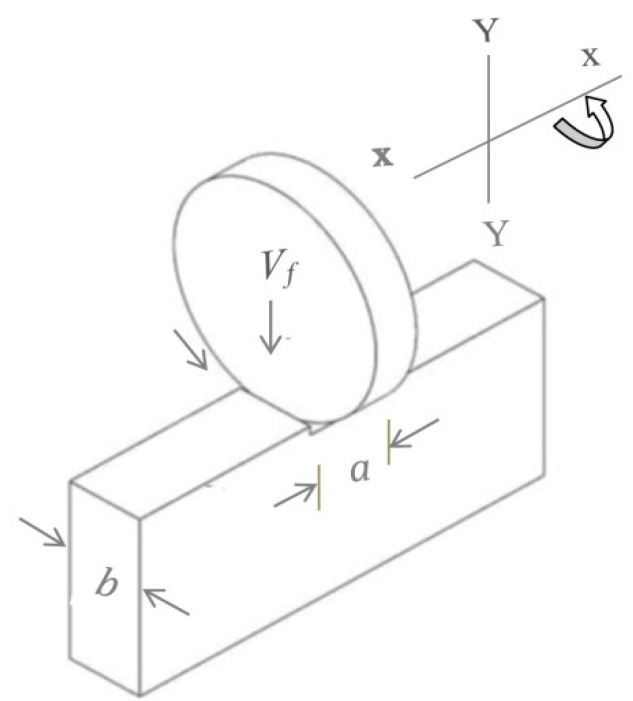
Cutting cross section.

**Figure 7 sensors-22-07152-f007:**
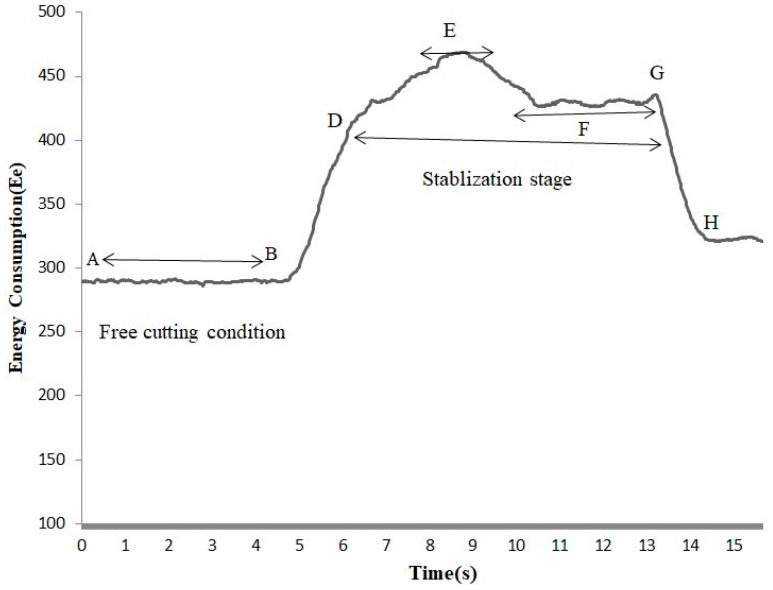
Energy consumption behavior during cutting with disc.

**Figure 8 sensors-22-07152-f008:**
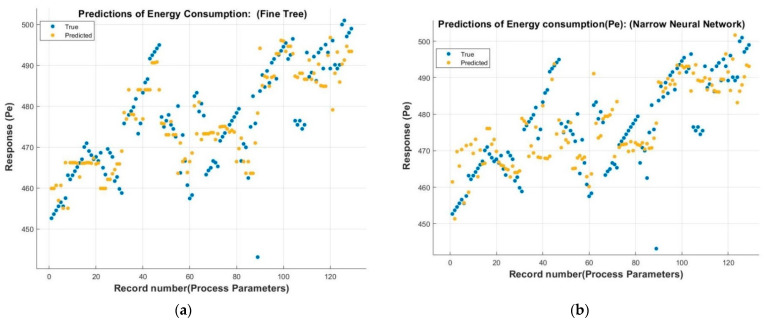
Predicted response of energy consumption (*E_e_*) with (**a**) regression trees, (**b**) Gaussian process regression, (**c**) ANN.

**Figure 9 sensors-22-07152-f009:**
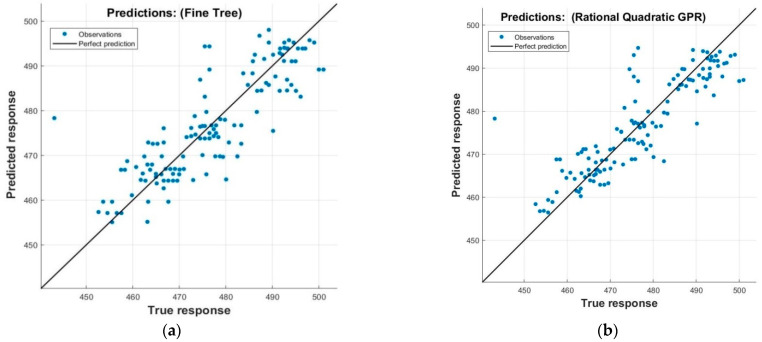
True vs predicted response. (**a**) GPR, (**b**) regression trees, (**c**) ANN.

**Figure 10 sensors-22-07152-f010:**
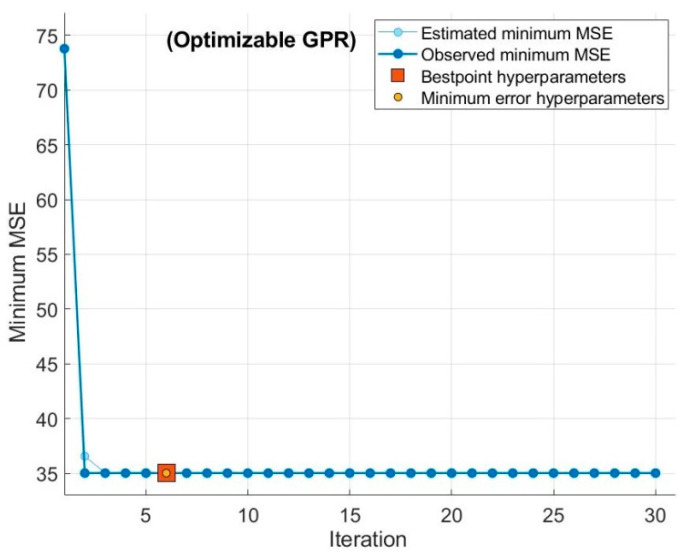
Optimization of Gaussian process regression (GPR).

**Figure 11 sensors-22-07152-f011:**
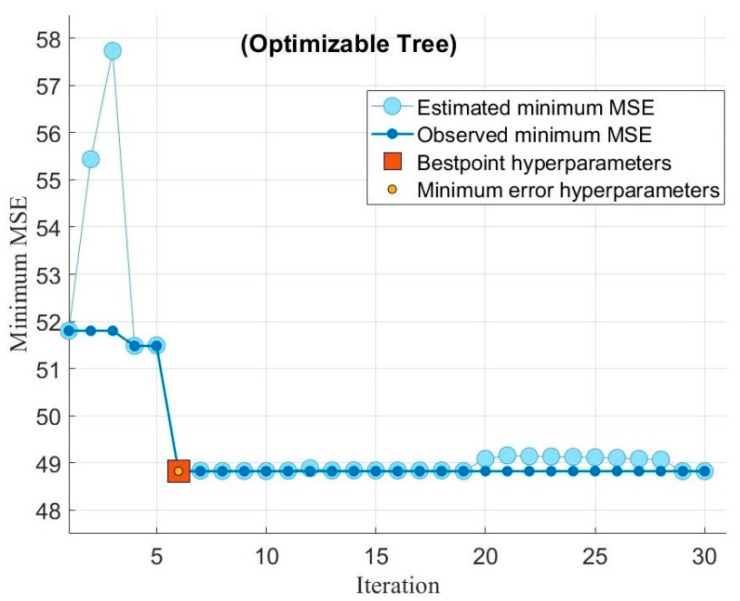
Optimization of regression trees.

**Figure 12 sensors-22-07152-f012:**
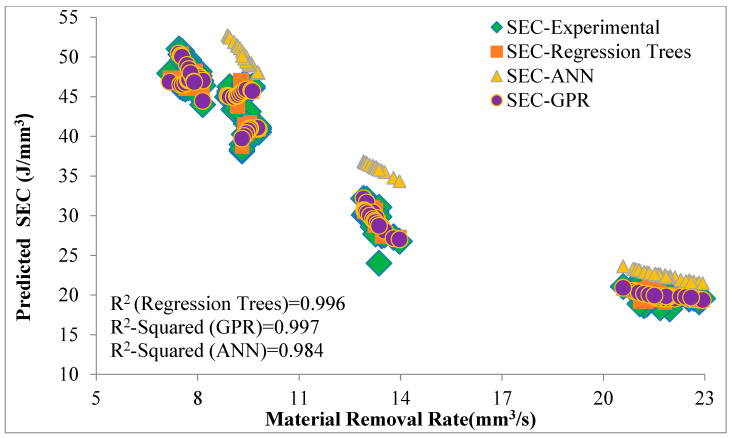
Validation of predicted SEC.

**Table 1 sensors-22-07152-t001:** Experimental cutting conditions.

Material	99.99% pure oxygen-free copper (OFC–C10100)
Cutting tool dimensions	1 × 115 × 22 mm with a maximum speed of 80 m/s
Feed rate (mm/s)	0.5–1.5 mm/s. Feed rate range can be changed.
Cutting thickness (a) Material thickness (b)	1.3–1.5 mm 10.8 mm

## Data Availability

Data will be deposited in public repositories.

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
