# Peer review of "Machine Learning-Based Prediction of Specific Energy Consumption for Cut-Off Grinding"

_sensors, 2022, doi:10.3390/s22197152_

Round 1
Reviewer 1 Report
The author makes a prediction study on grinding specific energy by using machine learning. The comments of the article are as follows:
1. There are too many keywords.
2. The format of "Fig 1" in row 107 is different from that of "Fig.2" in row 108. Please unify the format here. Other similar places in the article also need to be revised, such as row 155 and so on.
3. The measurement of electrical energy consumption (Ee) and mechanical cutting power (Pm) using Hystersis dynamomter (HD-710-BNA) is mentioned in section 2.2. The Digital microscope [WADEO iT33-MDUK] is also used to measure the cutting thickness at different point of the cutting path. However, the key parameters of these two sets of equipment are not included. And the unit of test results is not given.
4. Part 2.4 of the article can be merged into Part 2.3.
5. Three methods (Gaussain Process Regression, Regression Tress and Artistic Neural Network (ANN)) are used, and finally four output parameters (root mean square error (RMSE), determination coefficient R, mean square error (MSE), and mean absolute error (MAE)) are discussed. There are no interrelated conclusions or discussions in the discussion section of the article. In addition, please add the experimental pictures to the article.
6. The rotation speed of the grinding wheel and the relationship between the rotation direction of the grinding wheel and the moving direction of the workpiece are missing in the grinding parameters in Table 1.
7. “Table 1” should be at the top of the table in the article, and should be consistent with the table form in the “Appendix”.
Author Response
The reply has been uploaded as a file. I have revised all the sections in the manuscript according to your suggestions.

Reviewer 2 Report
This paper's study idea, which compares three theories and then identifies the one with the least mistake, is really sound. But the overall lack of innovation and insufficient depth of the research make me doubt that the manuscript can satisfy the journal's requirements.
Please see the details from attachment.

Author Response
My response to the feed back has been uploaded as a file. I have tried to address the possible changes following your recommendations. All the sections have been improved.

Reviewer 3 Report
Dear Authors,
my comments are as follows:
The Introduction Section should be improved, regarding the process monitoring and grinding technology.
Section Two - is 2.0, should be 2. , 3 not 3.0.
l. 108 consumption.First - space after "dot" is missing
The abrasive metal cutting is performed by 1x115 mm cutting discs, standard grinder - the detailed specifications of cutting discs and a grinder are missing.
Table 1: cutting thickness 1.3-1.8 mm: why there is so broad range? In appendix the thickness is from 1.3 to 1.5 mm?
There should be a space between the number and the unit, e.g. 1.5 mm, not 1.5mm.
Figure 6 must be improved according to standards of the technical drawing. The feed rate and rotations must be specified on the drawing.
Figure 6 is the same as the Figure 4 from [5].
This is not clear how the experiments were performed. 130 experiments, but according to what kind of plan? There are fewer experiments in the appendix.
What about the wear of the tool. When were the tools changed?
Where the experiments repeated several times?
The type of grinder used in the experiments is rather used for manual operations. The practical implications should be mentioned in the Conclusions.
Regards
Reviewer
Author Response
Many thanks for your time to review the manuscript. I have made the requested changes and revised the sections according to your recommendations. The point by point reply to your feed back has been uploaded as the file.

Round 2
Reviewer 2 Report
The revised paper has more tightly focused central content and has added pertinent information where it was previously missing. I concur that the paper is prepared for publication. But I wish the author had continued to make more attempts to increase the scientific depth of the article.
Author Response
Thank you for the time to review the article again. Yes, I understand your concern about the scientific depth. With this experimental setup and cutting conditions, this was the first attempt towards data-driven modeling, which provided us with sufficient information and constraints for data acquisition and modeling. In the future, using this information, more experimentation is planned to be performed with other materials like aluminum, steel, and Inconel. Which will provide more room for comparison of machine learning model performance among different materials. In addition, data cleaning and preprocessing will also be improved, and in depth modeling analysis will also be included. Doing these steps, I think the future manuscript in this direction will have more quality.
Reviewer 3 Report
Dear Authors,
I accept most of your modifications, but Figure 7 must be corrected. There is no axis of rotation, the size of the tool is disproportionate, and the direction of the rotations is not shown.
Regards
Reviewer
Author Response
Thank you very much for sparing precious time to review the article again. The figure has been corrected, axis and direction of rotation have been specified on the diagram.